# Effects of intravesical BCG maintenance therapy duration on recurrence rate in high-risk non-muscle invasive bladder cancer (NMIBC): Systematic review and network meta-analysis according to EAU COVID-19 recommendations

Young Joon Moon[1], Kang Su Cho[2], Jae Yong Jeong[1], Doo Yong Chung[3], Dong Hyuk Kang[3], Hae Do Jung[4], Joo Yong Lee[1,5]*

1 Department of Urology, Severance Hospital, Urological Science Institute, Yonsei University College of Medicine, Seoul, Korea, 2 Department of Urology, Gangnam Severance Hospital, Urological Science Institute, Yonsei University College of Medicine, Seoul, Korea, 3 Department of Urology, Inha University College of Medicine, Incheon, Korea, 4 Department of Urology, Inje University Ilsan Paik Hospital, Inje University College of Medicine, Goyang, Korea, 5 Center of Evidence Based Medicine, Institute of Convergence Science, Yonsei University, Seoul, Korea

* joouro@yuhs.ac

**Data Availability Statement:** All relevant data are within the paper, Table 1 and Fig 6.

## Abstract

### Purpose

During the coronavirus disease 2019 (COVID-19) pandemic, the European Association of Urology (EAU) recommended that courses of intravesical bacillus Calmette-Guérin (BCG) therapy lasting more than 1 year could be safely terminated for patients with high-risk non-muscle-invasive bladder cancer (NMIBC). Thus, we conducted a systematic review and network meta-analysis according to EAU's COVID-19 recommendations.

### Materials and methods

A systematic review was performed following the Preferred Reporting Items for Systematic Review and Meta-Analysis guidelines. We conducted a network meta-analysis of recurrence rate in patients with NMIBC receiving induction therapy (M0) and those receiving maintenance therapy lasting 1 year (M1) and more than 1 year (M2).

### Results

Nineteen studies of 3,957 patients were included for the network meta-analysis. In a node-split forest plot using Bayesian Markov Chain Monte Carlo (MCMC) modeling, there were no differences between the M1 and M2 groups in recurrence rate [odds ratio (OR) 0.95 (0.73–1.2)]. However, recurrence rate in the M0 group was higher than that in the M1 [OR 1.9 (1.5–2.5)] and M2 [OR 2.0 (1.7–2.4)] groups. P-score tests using frequentist inference to rank the treatments in the network demonstrated that the therapy used in the M2 group (P-

**Funding:** The author(s) received no specific funding for this work.

**Competing interests:** The authors have declared that no competing interests exist.

score 0.8701) was superior to that used in the M1 (P-score 0.6299) and M0 groups (P-score 0). In rank-probability tests using MCMC modeling, the M2 group showed the highest rank, followed by the M1 and M0 groups.

## Conclusion

In the network meta-analysis, there were no differences between those receiving BCG maintenance therapies in terms of recurrence rate. In the rank tests, therapy lasting more than 1-year appears to be most effective. During the COVID-19 pandemic, 1-year maintenance therapy can be used, but after the COVID-19 pandemic, therapy lasting more than 1-year could be beneficial.

## Introduction

Bladder cancer is the 10th most common cancer in the world. The incidence of bladder cancer is increasing globally [1]. Approximately 75–85% of patients have the non-muscle invasive bladder cancer (NMIBC) type [2]. The treatment of choice for NMIBC is transurethral resection of the bladder tumor (TURBT) [3]. Features of NMIBC include a high recurrence rate after TURBT and the potential risk of progression to muscle-invasive disease [4]. Bacillus Calmette-Guérin (BCG) immunotherapy is considered the most effective adjuvant treatment to prevent recurrence and progression of high-risk NMIBC after TURBT [5].

BCG is a vaccine against tuberculosis that has been used as an immunotherapy for bladder cancer for more than 40 years [6]. In particular, three-quarters of early diagnosed bladder cancers are NMIBC, which is characterized by a high recurrence rate. Reduction of disease recurrence and prevention of progression to muscle-invasive disease are important considerations in NMIBC management [4]. In this aspect, intravesical BCG immunotherapy has been used as the backbone of adjuvant therapy after TURBT in patients with NMIBC [7].

Although the European Association of Urology (EAU)'s guidelines recommend 3-week instillations at 3, 6, 12, 18, 24, 30, and 36 months based on European Organization for Research and Treatment of Cancer data, the optimal duration of maintenance BCG is still unknown [8].

The coronavirus disease 2019, also known as COVID-19, has spread around the world, and the World Health Organization officially declared it a pandemic on March 11, 2020. Health-care has been severely impacted, and urology practices have also been affected by the COVID-19 pandemic [9]. According to EAU's COVID-19 recommendations, EAU's NMIBC panel recommends 1-year intravesical BCG maintenance immunotherapy in patients with high-risk NMIBC [10]. Therefore, the purpose of this study was to determine the appropriate duration of BCG maintenance therapy during the COVID-19 pandemic since 2020.

## Materials and methods

### Inclusion criteria

We defined study eligibility following the Preferred Reporting Items for Systematic Review and Meta-Analysis (PRISMA) guidelines (S1 Table) [11]. The patient population was histologically confirmed to be NMIBC positive in tissue collected after TURBT. The intervention was intravesical BCG immunotherapy. The comparator was duration. The outcome was recurrence rate. The study design was a systematic review and meta-analysis. Patients were categorized into three groups: induction BCG therapy only (M0), 1-year BCG maintenance therapy

(M1), and BCG maintenance therapy lasting more than 1 year (M2). Patients in the BCG maintenance therapy groups (M1 and M2) received induction BCG therapy followed by regular BCG maintenance therapy for at least 1 year, while patients in the M0 group received induction BCG therapy only. Any strain or dose of BCG was considered appropriate. We conducted this study based on the standard PRISMA guidelines [12].

## Search strategy

Literature searches for all publications prior to September 31, 2021 were carried out using PubMed and EMBASE. The following medical subject headings terms and keywords were used for the search: "urinary bladder neoplasms," "urothelial carcinoma of bladder," "transitional cell carcinoma of bladder,""bladder carcinoma," "bladder cancer," "BCG," "Bacillus Calmette-Guérin," and "maintenance."

## Data extraction

Two researchers (YJM and JYJ) screened the titles and abstracts of articles that were independently identified by the search strategy to exclude irrelevant studies. They also evaluated the full text of the articles to find potentially related articles. They extracted the most relevant articles in each study. Disagreements were solved by debate among the researchers until a consensus was reached.

## Quality assessment for studies

In the case of randomized controlled trials (RCTs), the Cochrane Risk of Bias tool was used, and in the case of nonrandomized studies, the methodological index for nonrandomized studies (MINORS) was used. Quality of evidence grading was performed using the Scottish Intercollegiate Guidelines Network (SIGN) checklist, consisting of various types of research. The quality assessment was conducted independently by our researchers.

## Heterogeneity tests

The Q statistic and Higgins' $I^2$ statistic were used for evaluations of study heterogeneity. Higgins' $I^2$ was calculated as follows:

$$I^2 = \frac{Q - df}{Q} \times 100\%$$

where "Q" is Cochran's heterogeneity statistic and "df" is the degrees of freedom. When the *P* value was less than 0.10, heterogeneity was considered significant. If evidence of heterogeneity existed, the data were analyzed using a random-effects model. Studies in which positive results had been confirmed were assessed with a pooled specificity using 95% confidence intervals (CIs).

## Statistical analysis

The primary outcome was tumor recurrence, which was measured using the odds ratio (OR) with 95% CIs. All statistical analyses were performed with R software (version 4.1.2, R Foundation for Statistical Computing, Vienna, Austria; http://www.r-project.org) and with the associated meta, netmeta, pcnetmeta, and gemtc packages for pairwise and network meta-analyses. This systematic review is registered in PROSPERO, CRD 42021291265.

## Results

### Eligible studies

A total of 1,602 articles were verified by the initial database search. Of these, 1,409 articles were excluded: 502 were duplicate publications and 907 were excluded after reviewing the abstracts. A total of 193 articles were selected for full text evaluation. Further review excluded 174 articles because they were not relevant to the analysis: 89 were out of scope; 15 had a questionable study design; 39 were in review journals; 25 cited improper interventions; and 6 were excluded due to other causes. Finally, 19 articles were selected for the meta-analysis. Fig 1 shows the study flow chart.

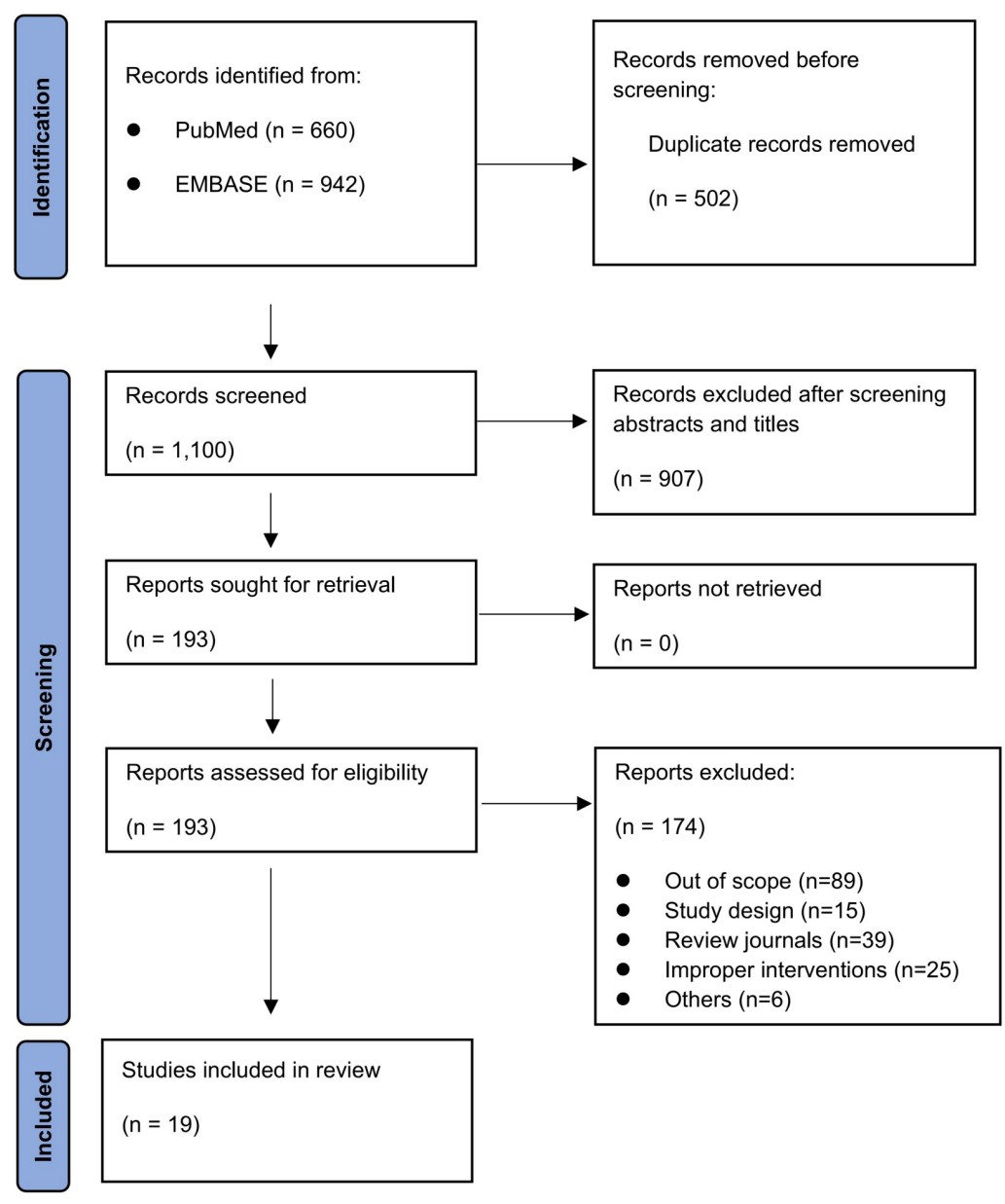

**Fig 1. Study flow chart.**

## Characteristics of included studies with quality assessment and publication bias

The characteristics and recurrence rates described in the 19 included studies are shown in Table 1 [13–31]. These eligible studies were published between 1987 and 2021. Nineteen studies with a total of 3,957 patients were included in the qualitative and quantitative analyses. There were just two published studies that included the M1 and M2 groups. Five studies included the M0 and M1 groups, and 12 studies included the M0 and M2 groups (Fig 2).

The quality assessment results using SIGN are provided in Table 1. Funnel plots of our study are shown in Fig 3. Most studies were located in the funnels. The risk of bias for eight RCTs is displayed in Figs 4 and 5. Adequate randomization methods and allocation concealment were described in only six and three studies, respectively. Blinding of outcome assessments was performed in five studies. The MINORS scores are shown in Table 2. All studies were considered appropriate.

## Heterogeneity and inconsistency assessment

Forest plots of the pairwise meta-analysis results of the three groups are shown in Fig 6. There was no heterogeneity between groups M1 and M2 or between groups M0 and M2 in any study; however, there was little heterogeneity between groups M0 and M1. Therefore, a random-effects model was applied for a comparison of groups M0 and M1 (Fig 6B). After selection of the random-effects model, little heterogeneity was noted in L'Abbe plots (Fig 7) and radial plots (Fig 8). In the node-splitting analysis, no inconsistency was demonstrated in direct, indirect, or network comparisons (Fig 9).

## Pairwise meta-analysis of groups M0, M1, and M2

The recurrence rate of the M1 group was slightly higher than that of the M2 group ($P = 0.328$; OR 1.161, 95% CI 0.861–1.564) (Fig 6A). The recurrence rate of the M0 group was higher than that of the M1 group ($P = 0.013$; OR 2.877, 95% CI 1.246–6.643) (Fig 6B). The recurrence rate of the M0 group was also higher than that of the M2 group ($P < 0.001$; OR 1.958, 95% CI 1.618–2.369) (Fig 6C).

## Network meta-analysis of groups M0, M1, and M2 for recurrence rate

In the node-split forest plot using Bayesian Markov Chain Monte Carlo (MCMC) modeling, there were no differences between the M1 and M2 groups in terms of recurrence rate [OR 0.95 (0.73–1.2)]. However, the recurrence rate in the M0 group was higher than those in the M1 [OR 1.9 (1.5–2.5)] and M2 [OR 2.0 (1.7–2.4)] groups (Fig 9). P-score tests using frequentist inference to rank treatments in the network demonstrated that the M2 group treatment (P-score 0.8701) was superior to the M1 (P-score 0.6299) and M0 groups (P-score 0). In the rank-probability test using MCMC modeling, the M2 group showed the highest rank, followed by the M1 and M0 groups (Fig 10).

## Discussion

BCG was first discovered by Albert Calmette and Camille Guerin as a tuberculosis vaccination in 1921 [32]. The first report of BCG as an immunotherapy for bladder cancer was published in 1959 [33]. In 1976, Morales et al. published a landmark paper about the beneficial effects of BCG immunotherapy on recurrent superficial bladder cancer [34]. In 1980, the first controlled trial showing similar results was published, and in 1990, BCG received Food and Drug Administration approval for the treatment of superficial bladder cancer [35].

**Table 1. Characteristics of included studies.** M0, induction BCG therapy only; M1, 1-year BCG maintenance therapy; M2, BCG maintenance therapy for more than 1 year.

| Category | Study | Methods | Study design | BCG strain | No. of Patients | Mean/median follow-up period | Recurrence | Quality assessment (SIGN) |
|---|---|---|---|---|---|---|---|---|
| M0 vs. M1 | Mohamed et al. 2020 [13] | M0 | Prospective | Pasteur | 27 | 40 months | 15 | 2+ |
| | | M1 | | | 26 | 35 months | 5 | |
| | Yoo et al. 2012 [14] | M0 | Retrospective | OncoTICE | 34 | 16.5 months | 16[a] | 1+ |
| | | M1 | | | 92 | 43 months | 21[a] | |
| | Okamura et al. 2011 [15] | M0 | Retrospective | Tokyo | 27 | 66 months | 13[b] | 1+ |
| | | M1 | | | 48 | 102 months | 8[b] | |
| | Koga et al. 2010 [16] | M0 | Randomized controlled | Tokyo | 27 | 28.7 months | 7[a] | 2+ |
| | | M1 | | | 24 | 26.5 months | 1[a] | |
| | Akaza et al. 1995 [17] | M0 | Randomized controlled | Tokyo | 55 | 42 months | 20 | 1+ |
| | | M1 | | | 52 | 48 months | 22 | |
| M0 vs. M2 | Miyake et al. 2021 [18] | M0 | Retrospective | Tokyo or Connaught | 874 | 48 months | 175[a] | 2+ |
| | | M2 | | | 405 | | 41[a] | |
| | Koguchi et al. 2020 [19] | M0 | Retrospective | Tokyo | 40 | 36.2 months | 14 | 2+ |
| | | M2 | | | 38 | | 5 | |
| | Joshua et al. 2019 [20] | M0 | Retrospective | Not addressed | 40 | Not addressed | 8[a] | 1+ |
| | | M2 | | | 61 | | 7[a] | |
| | Yuk et al. 2018 [21] | M0 | Retrospective | Not addressed | 29 | 63 months | 14 | 2+ |
| | | M2 | | | 26 | | 5 | |
| | Nakai et al. 2016 [22] | M0 | Randomized controlled | Connaught | 42 | 51 months | 9[b] | 2+ |
| | | M2 | | | 46 | | 9[b] | |
| | Martínez-Piñeiro et al. 2015 [23] | M0 | Randomized controlled | Connaught | 195 | 103 months | 80[b] | 2+ |
| | | M2 | | | 202 | 102 months | 68[b] | |
| | Muto et al. 2013 [24] | M0 | Retrospective | Connaught | 64 | 42.3±33.1 months | 23[b] | 1+ |
| | | M2 | | | 40 | 51.1±34.6 months | 6[b] | |
| | Hinotsu et al. 2011 [25] | M0 | Randomized controlled | Connaught | 42 | Not addressed | 14[a] | 1+ |
| | | M2 | | | 41 | | 5[a] | |
| | Palou et al. 2001 [26] | M0 | Randomized controlled | Connaught | 61 | 77.8 months | 16 | 2+ |
| | | M2 | | | 65 | | 10 | |
| | Lamm et al. 2000 [27] | M0 | Randomized controlled | Connaught | 192 | Not addressed | 113[b] | 2+ |
| | | M2 | | | 192 | | 77[b] | |
| | Badalament et al. 1987 [28] | M0 | Randomized controlled | Pasteur | 46 | 22 months | 3 | 1+ |
| | | M2 | | | 47 | | 6 | |
| | Hudson et al. 1987 [29] | M0 | Randomized controlled | Pasteur | 21 | 17.2 months | 6 | 1+ |
| | | M2 | | | 21 | | 5 | |
| M1 vs. M2 | Gupta et al. 2020 [30] | M1 | Randomized controlled | Moscow | 38 | Not addressed | 5[b] | 2+ |
| | | M2 | | | 40 | | 6[b] | |
| | Oddens et al. 2013 [31] | M1 | Randomized controlled | OncoTICE | 339 | 7.1 years | 145 | 2+ |
| | | M2 | | | 338 | | 131 | |

[a]During the 2-year follow-up;
[b]during the 5-year follow-up.
The quality assessment was indicated by Scottish Intercollegiate Guidelines Network (SIGN) checklist. 1+ means well-conducted RCT with a low risk of bias. 1- means RCT with a high risk of bias. 2+ means well-conducted cohort studies with a low risk of bias. 2- means cohort studies with a high risk of bias.

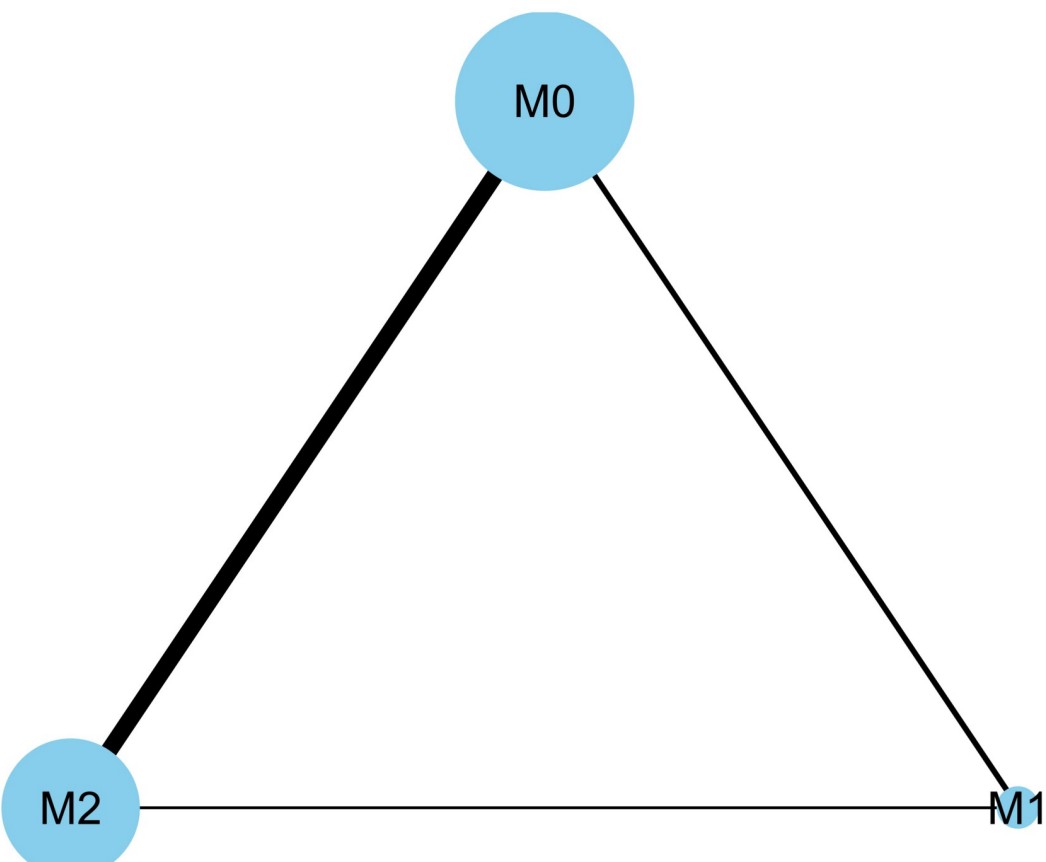

**Fig 2. Network plots for included studies.** There were just two published studies that included the M1 and M2 groups. Five studies included the M0 and M1 groups, and 12 studies included the M0 and M2 groups. M0, induction BCG therapy only; M1, 1-year BCG maintenance therapy; M2, BCG maintenance therapy for more than 1 year.

Several studies have demonstrated that BCG maintenance treatments show clinical benefit in patients with high-risk NMIBC. Lamm et al. reported the effectiveness of BCG maintenance treatments in 384 patients with recurrent NMIBC [27]. Compared with standard induction BCG immunotherapy, BCG maintenance immunotherapy was favorable in patients with superficial bladder cancer. Compared to the induction therapy-only arm, patients in the 3-week maintenance arm showed twice as long median recurrence-free survival (RFS) and significantly longer progression-free survival. Mohamed et al. reported a prospective randomized study comparing 31 patients with NMIBC who underwent induction therapy only and 35 patients with NMIBC who underwent induction therapy plus a 1-year maintenance therapy [13]. Patients who received only induction therapy had significantly higher recurrence rates than those who received maintenance therapy. The 5-year RFS rate was 41% in the induction therapy group and 78% in the maintenance therapy group. To date, BCG maintenance treatments are considered to exert clinical benefits, especially in terms of preventing recurrence of NMIBC.

Although the optimal duration of maintenance therapy for patients with high-risk NMIBC remains controversial, EAU's guidelines recommend 3-week instillations at 3, 6, 12, 18, 24, 30, and 36 months. As the COVID-19 pandemic has had a serious impact on urological treatments [9], Lenfant et al. reported that intravesical BCG therapy could be discontinued safely for patients with high-risk NMIBC [8]. Furthermore, according to EAU's NMIBC panel during

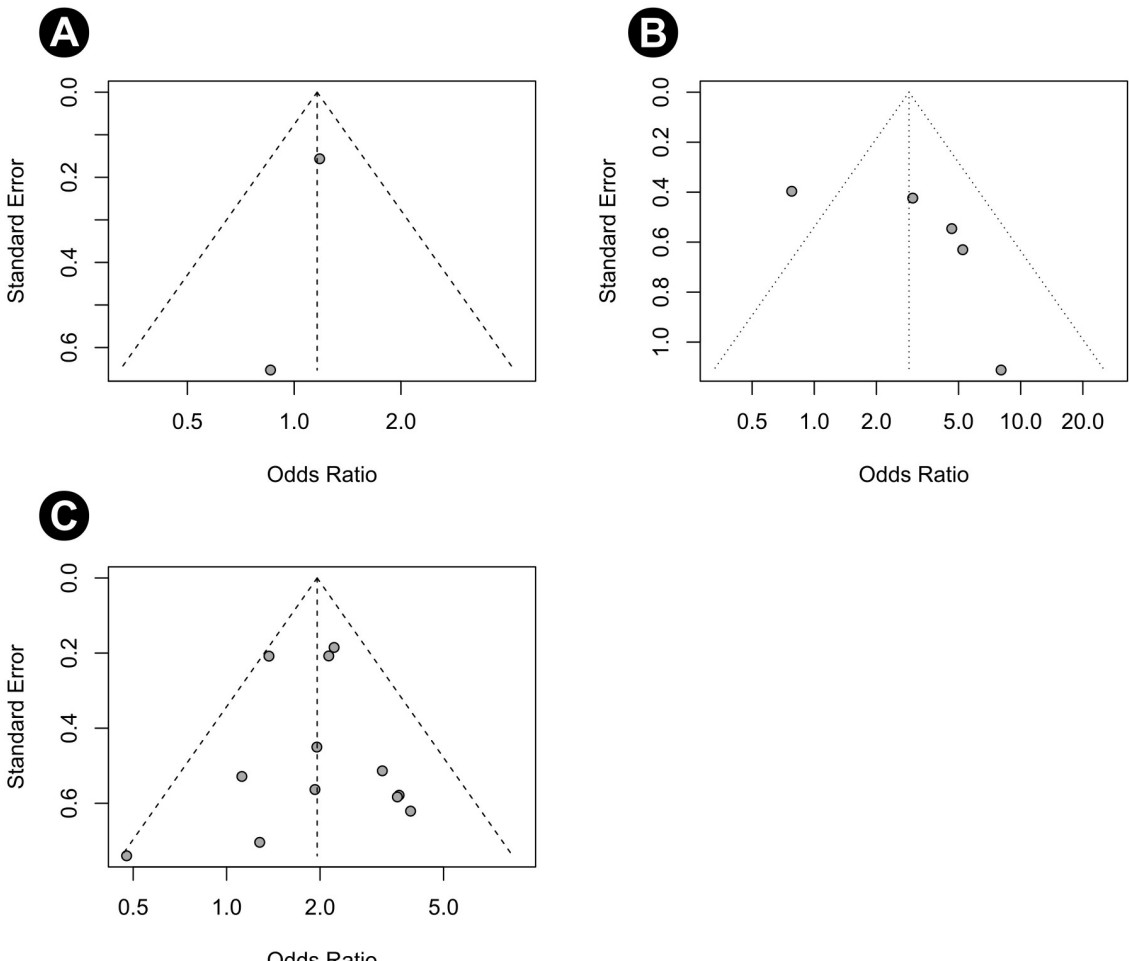

**Fig 3. Funnel plot.** (A) Recurrence rate in the M0 group, (B) recurrence rate in the M1 group, and (C) recurrence rate in the M2 group.

the pandemic, a 1-year BCG maintenance immunotherapy in patients with high-risk NMIBC was recommended [10].

There have been several systematic reviews and meta-analyses on intravesical BCG treatment in patients with NMIBC. Chen et al. conducted a systematic review and meta-analysis of 10 RCTs [36], showing that BCG maintenance therapy could decrease the risk of tumor recurrence by 21% and prolong RFS by 33% compared with nonmaintenance therapy. In addition, they showed that BCG maintenance therapy could decrease the risk of tumor progression. These results are the basis for the effectiveness of BCG maintenance treatment. Quan et al. conducted a study about dose, duration, and BCG strain for the treatment of patients with NMIBC [4]. Finally, 19 studies were selected for a meta-analysis. Low-dose BCG and induction therapy-only groups showed significantly higher risks of recurrence [risk ratio (RR) 1.17 and 1.33, respectively]. These results may serve as the basis for the better clinical outcomes of the 3-year maintenance therapy. Huang et al. conducted a systematic review and meta-analysis of nine RCTs [37]. Similar to our study results, longer BCG maintenance therapy (such as 3 years) did not significantly reduce the risk of tumor recurrence or progression compared to

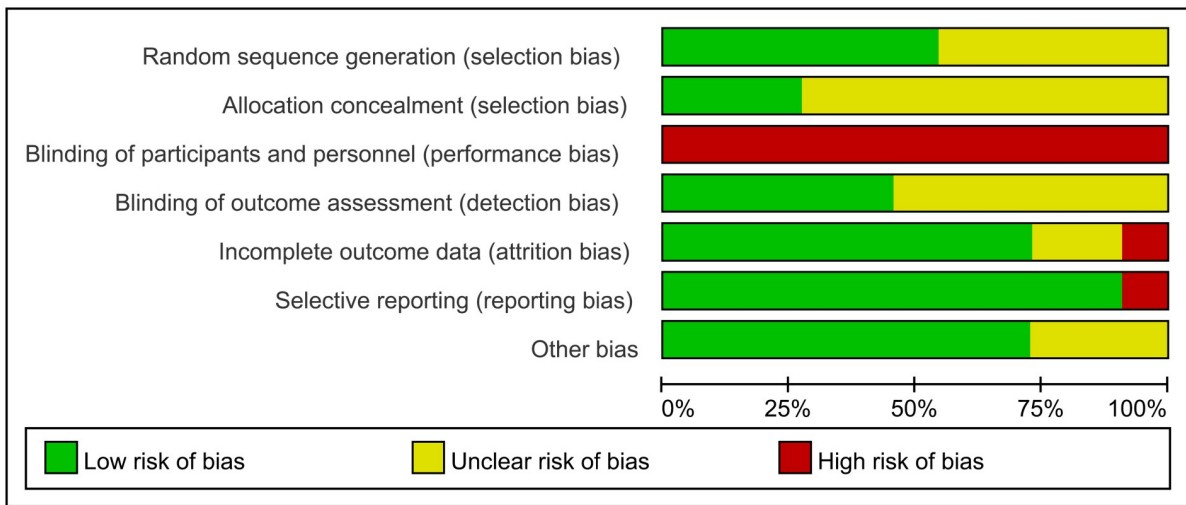

**Fig 4. Risk of bias for eight RCTs.** The risk of bias for each item is presented as a percentage across all included studies.

shorter-term BCG maintenance therapy (such as 1 year). However, the limitation is that all of these studies were conducted prior to the COVID-19 pandemic.

The characteristics of patients with NMIBC are generally a high median age of 70 years, many comorbidities, and a high smoking rate, which all increase the risk of COVID-19 severity [38]. Our study was conducted in consideration of these risks and EAU's COVID-19 recommendations; in the short term, there was no significant difference between maintenance therapy groups. Gallegos et al. conducted a prospective study on a total of 175 patients with NMIBC who received BCG treatment at a Chilean hospital from 2019 to 2020 [39]. Throughout the study duration, 43 patients were diagnosed with COVID-19. In these patients, only one patient died from the disease (case fatality rate = 2.3%) during follow-up. They also compared patients with COVID-19 receiving BCG treatment with the overall population of the same age (70–79 years), according to the Chilean national register. During the same follow-up duration, 6.3% of the control group became infected with COVID-19, with a 14% case fatality rate. With regard to the study results, patients with NMIBC receiving the BCG immunotherapy showed a lower case fatality rate than the control group, but a higher rate of COVID-19 infection. The cause of the high rate of infection in the BCG treatment group is unclear. However, since social distancing must be maintained to prevent COVID-19 infection, reducing the duration of maintenance treatment according to EAU's COVID-19 recommendations is a reasonable approach to take during the COVID-19 pandemic.

During the COVID-19 pandemic, management also has been delayed for muscle-invasive bladder cancer (MIBC). According to EAU recommendations, Kang et al. conducted systematic review and meta-analysis to evaluate the efficacy of neoadjuvant chemotherapy (NAC) compared with radical cystectomy (RC) alone in improving the overall survival (OS) of patients with T2-4aN0M0 MIBC [40]. The OS was significantly better in the NAC with RC group than in the RC alone group. However, in a subgroup analysis of patients with T2N0M0 MIBC, there was no difference in the OS between the NAC with RC group and the RC alone group. They concluded that, as recommended by the EAU Guidelines Office Rapid Reaction Group, patients with T2N0M0 MIBC should strongly consider omitting NAC until the end of the COVID-19 pandemic.

BCG unresponsiveness is one of the important considerations in BCG therapy for patients with NMIBC. According to the EAU guidelines on NMIBC, BCG unresponsive tumors

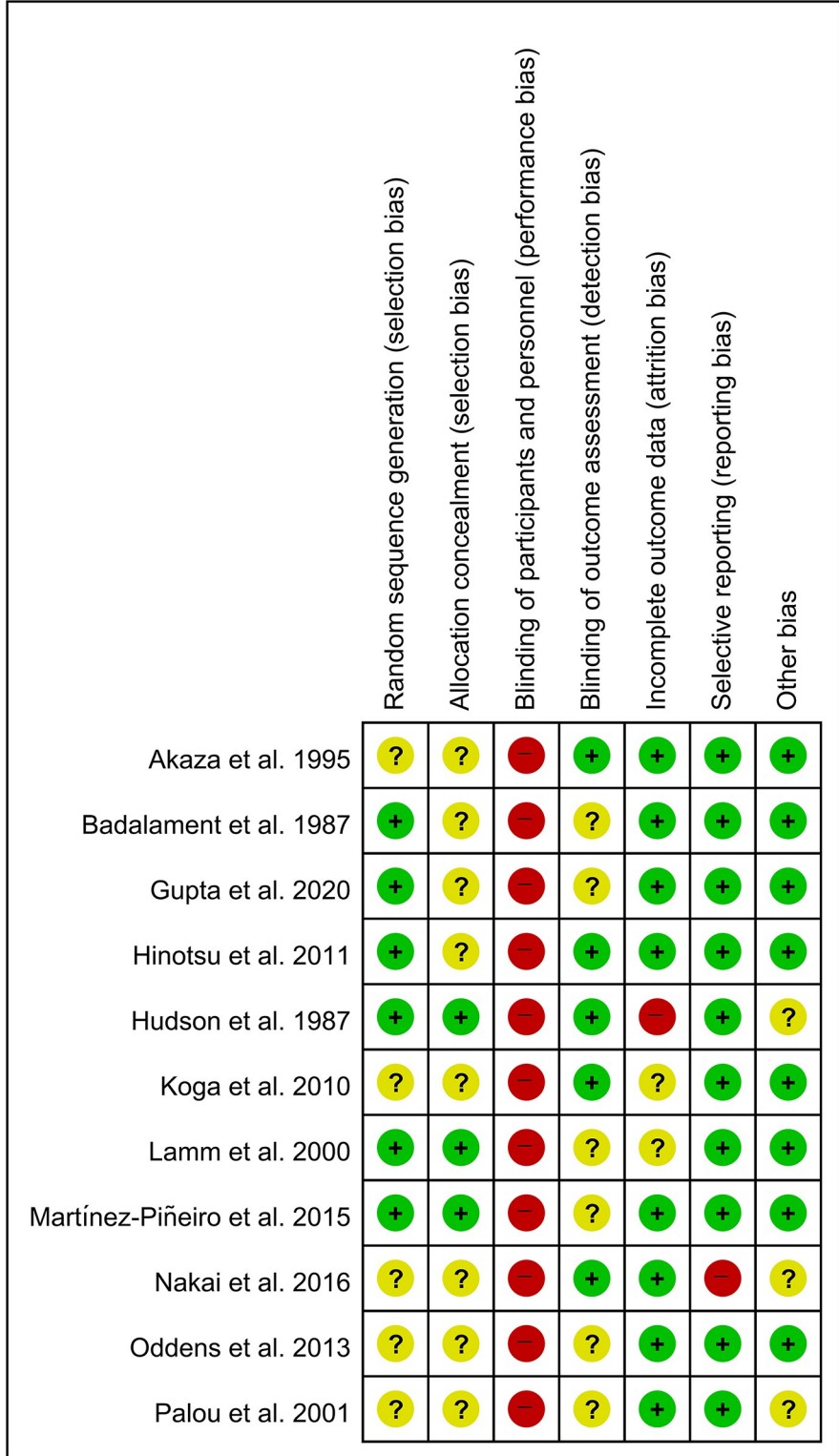

**Fig 5. Risk of bias for eight RCTs.** +, no bias;−, bias;?, bias unknown.

**Table 2. MINORS score in nonrandomized studies included in the review.**

| | A Clearly Stated Aim | Inclusion of Consecutive Samples | Prospective Collection of Data | Endpoints Appropriate to the Aim of the Study | Unbiased Assessment of the Study Endpoint | Follow-up Period Appropriate to the Aim of the Study | Loss to Follow-Up Less than 5% | Prospective Calculation of the Study Size | An Adequate Control Group | Contemporary Groups | Baseline Equivalence of Groups | Adequate Statistical Analyses | Total |
|---|---|---|---|---|---|---|---|---|---|---|---|---|---|
| Miyake et al. 2021 | 2 | 2 | 2 | 2 | 0 | 2 | 2 | 0 | 2 | 2 | 2 | 2 | 20 |
| Mohamed et al. 2020 | 2 | 2 | 2 | 2 | 0 | 2 | 2 | 0 | 2 | 2 | 1 | 2 | 19 |
| Koguchi et al. 2020 | 2 | 2 | 2 | 2 | 0 | 2 | 2 | 0 | 2 | 2 | 1 | 2 | 19 |
| Joshua et al. 2019 | 2 | 2 | 2 | 2 | 0 | 2 | 2 | 0 | 2 | 2 | 1 | 2 | 19 |
| Yuk et al. 2018 | 2 | 2 | 2 | 2 | 0 | 2 | 2 | 0 | 2 | 2 | 2 | 2 | 20 |
| Muto et al. 2013 | 2 | 2 | 2 | 2 | 0 | 2 | 2 | 0 | 2 | 2 | 1 | 2 | 19 |
| Yoo et al. 2012 | 2 | 2 | 2 | 2 | 0 | 2 | 2 | 0 | 2 | 2 | 2 | 2 | 20 |
| Okamura et al. 2011 | 2 | 2 | 2 | 2 | 0 | 2 | 2 | 0 | 2 | 2 | 2 | 2 | 20 |

MINORS, methodological index for nonrandomized studies. The items are scored 0 (not reported), 1 (reported but inadequate), or 2 (reported and adequate). The global ideal score is 16 for non-comparative studies and 24 for comparative studies.

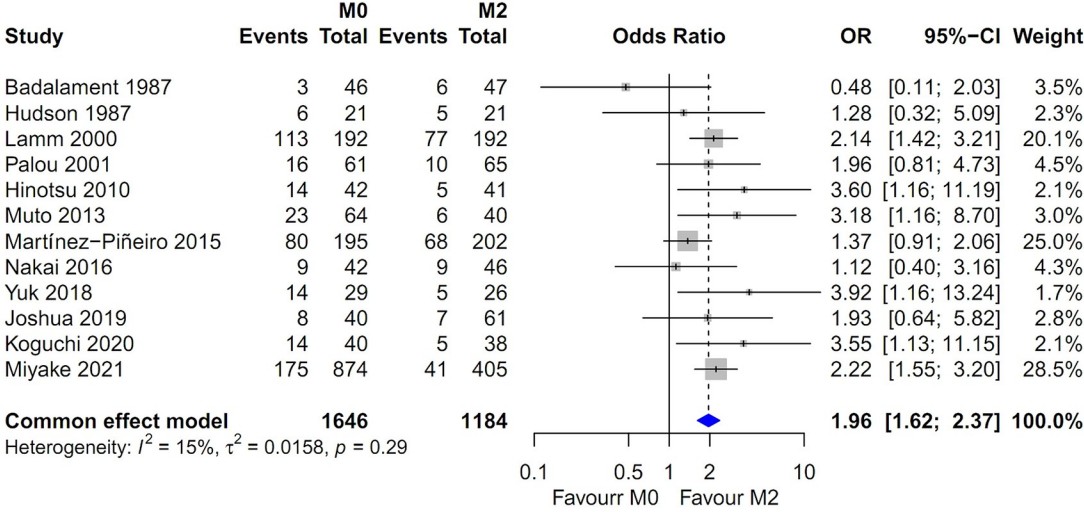

**Fig 6. Pairwise meta-analysis of (A) M1 and M2 groups, (B) M0 and M1 groups, and (C) M0 and M2 groups.** The recurrence rate in the M1 group was slightly higher than that in the M2 group ($P = 0.328$; OR 1.161, 95% CI 0.861–1.564). The recurrence rate in the M0 group was higher than that in the M1 ($P = 0.013$; OR 2.877, 95% CI 1.246–6.643) and M2 groups ($P < 0.001$; OR 1.958, 95% CI 1.618–2.369). M0, induction BCG therapy only; M1, 1-year BCG maintenance therapy; M2, BCG maintenance therapy for more than 1 year.

included all BCG refractory tumors (T1G3/high-grade (HG) tumor at 3 months; TaG3/HG tumor after 3 months and/or at 6 months, after either re-induction or first course of maintenance; carcinoma in situ (CIS), without concomitant papillary tumor, at 3 months and persisting at 6 months after either re-induction or first course of maintenance; HG tumor during BCG maintenance therapy) and those who develop T1Ta/HG recurrence within 6 months or CIS within 12 months from the completion of adequate BCG exposure [41]. In general, BCG-

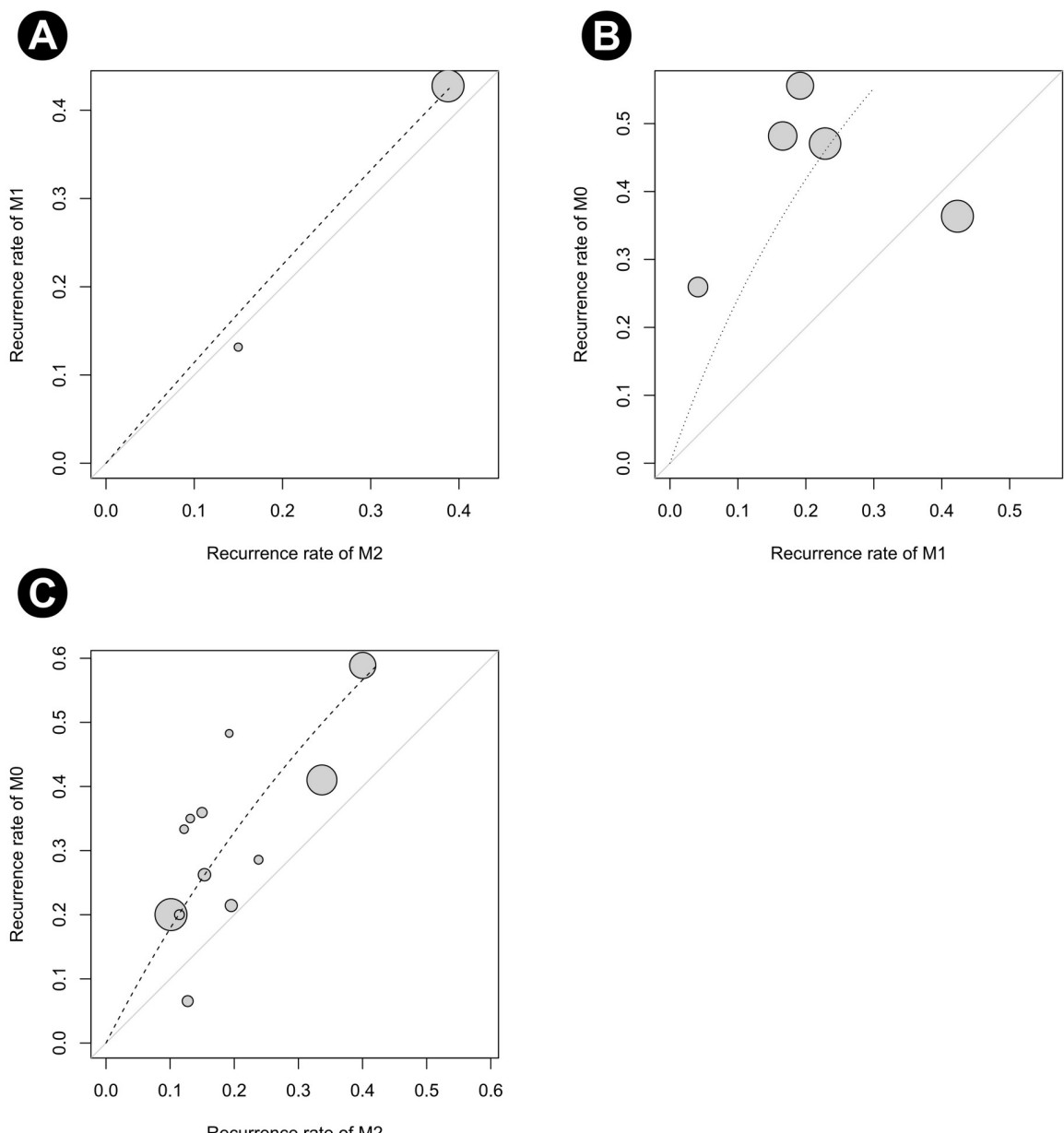

**Fig 7. L'Abbe plots of recurrence rate.** (A) Between the M1 and M2 groups, (B) between the M0 and M1 groups, and (C) between the M0 and M2 groups. M0, induction BCG therapy only; M1, 1-year BCG maintenance therapy; M2, BCG maintenance therapy for more than 1 year.

unresponsive patients have worse oncological outcomes, and therefore, studies related to factors that can predict BCG response are important in the treatment of NMIBC. Ferro *et al.* conducted a retrospective study to investigate the predictive factors in the response to BCG in patients with a T1G3/HG NMIBC diagnosis [42]. According to their study, multifocality, lymphovascular invasion, and HG on re-TURBT were independent predictors for response to BCG treatment. To reduce the risk of understaging and missing MIBC, re-TURBT should be performed, especially in HG NMIBC. According to another study, independent predictors to

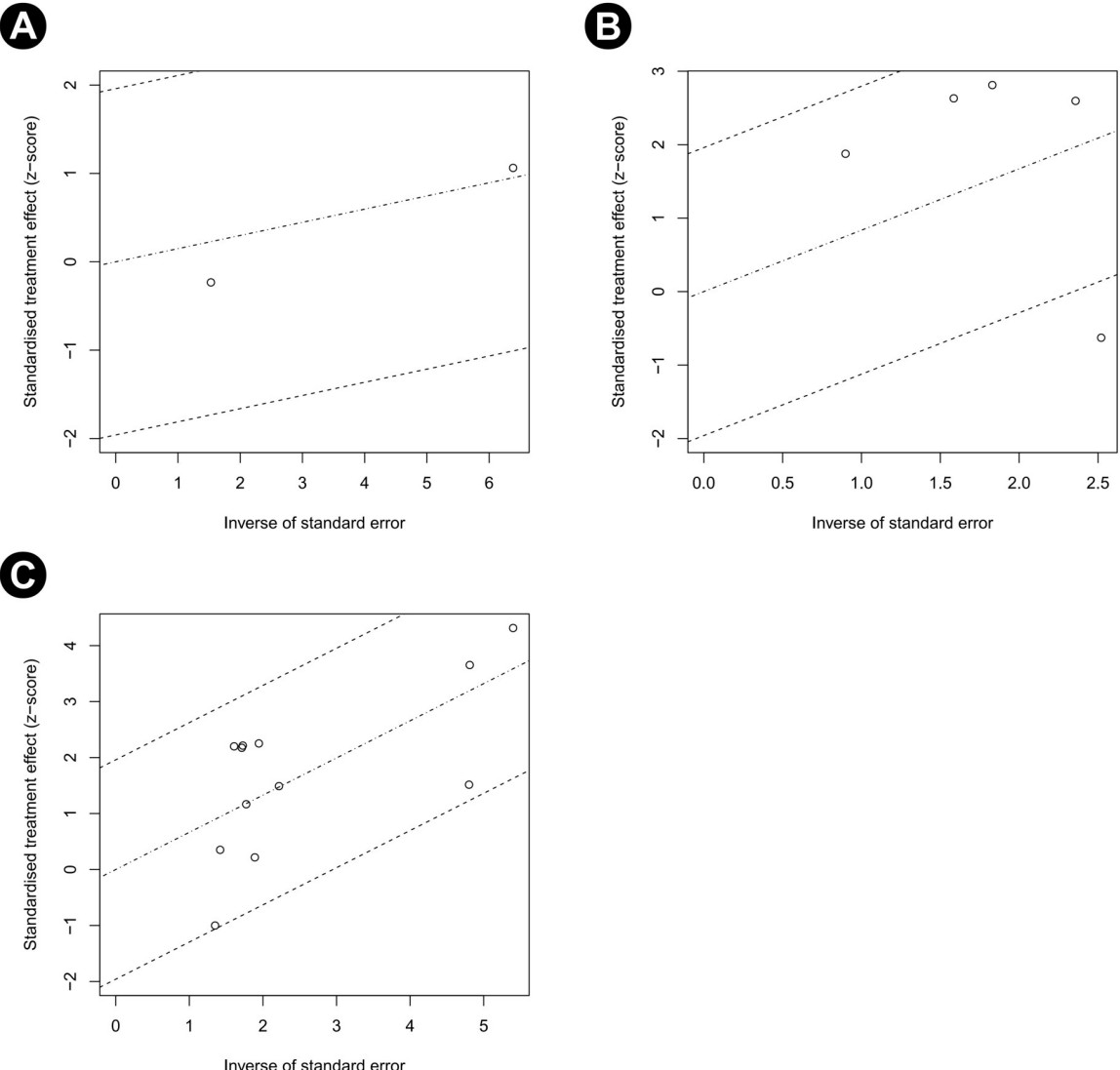

**Fig 8. Radial plots of recurrence rate.** (A) Between the M1 and M2 groups, (B) between the M0 and M1 groups, and (C) between the M0 and M2 groups. M0, induction BCG therapy only; M1, 1-year BCG maintenance therapy; M2, BCG maintenance therapy for more than 1 year.

identify patients at risk of residual HG disease after a complete TURBT include the tumor size, presence of CIS, and body mass index (BMI)≥25 kg/m$^2$ [43]. When deciding to perform re-TURBT, the presence of these factors would also be an important consideration.

The main strength of our study is that it is the first to evaluate differences in clinical outcomes with regard to the period of intravesical BCG maintenance therapy for patients with NMIBC during the era of COVID-19. Considering the COVID-19 infection rate, the fatality rate in patients with NMIBC, and the increase in the number of COVID-19-related deaths, it is reasonable to follow EAU's COVID-19 recommendations in the post-COVID-19 era. However, if COVID-19 becomes a controllable disease, a conventional BCG maintenance therapy can help decrease the recurrence rate of NMIBC. According to EAU's COVID-19

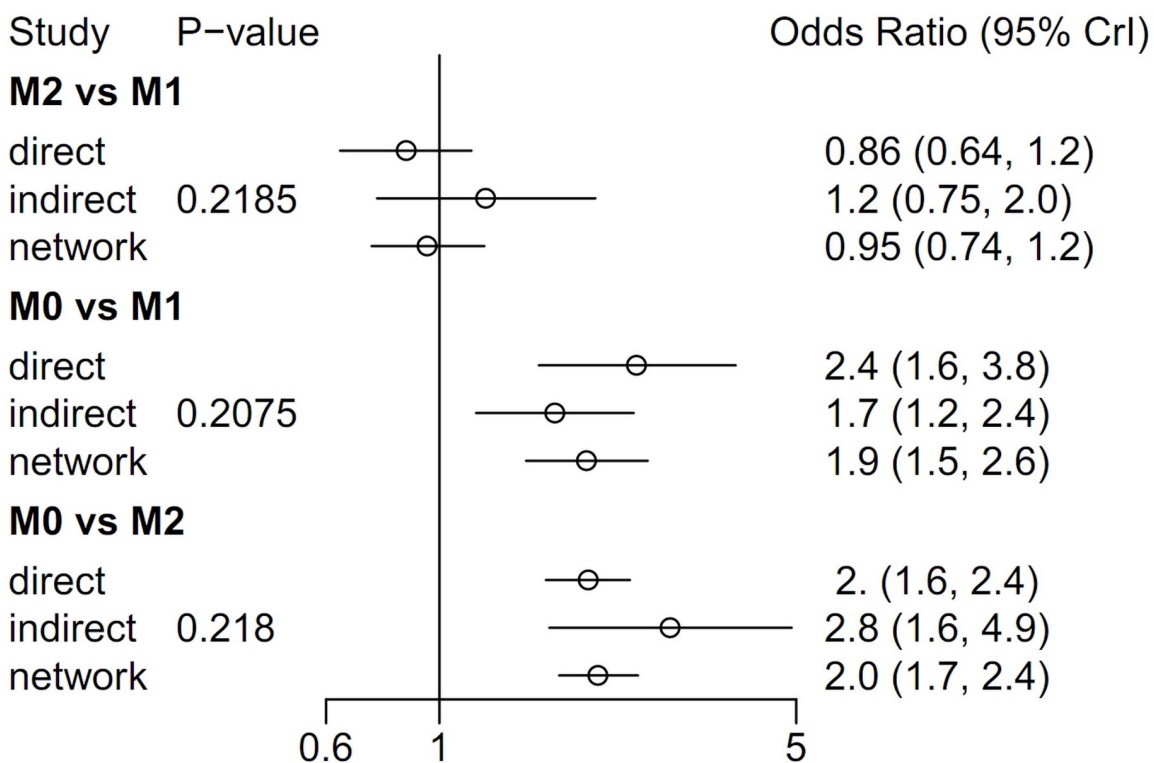

**Fig 9. Node-split forest plot using MCMC modeling.** M0, induction BCG therapy only; M1, 1-year BCG maintenance therapy; M2, BCG maintenance therapy for more than 1 year.

recommendations, studies on patients receiving BCG maintenance therapy for 1 year should be conducted in the near future, and preparations for another pandemic should be made.

The main limitation of our meta-analysis is its scant heterogeneity across the studies in terms of different treatment regimens and different strains of BCG used. Second, RCTs and non-RCT studies were mixed and analyzed in our network meta-analysis. Third, other clinical outcomes, such as progression rate and survival rate, were not analyzed. Further analysis of progression and survival rates is expected to increase the reliability of our results. Fourth, the side effects of intravesical BCG maintenance therapy and subsequent treatment tolerance were not analyzed. The severity of side effects is a significant factor in deciding on the duration of intravesical BCG therapy.

## Conclusions

In our network meta-analysis, there was no difference between BCG maintenance therapy groups in terms of recurrence rate. In the rank test, BCG therapy lasting more than 1-year appears to be most effective in patients with NMIBC.

Given the COVID-19 infection rate, fatality rate of NMIBC, and increase in the number of COVID-19 deaths, it is reasonable to follow the EAU COVID-19 recommendation for the post-COVID-19 era. However, if COVID-19 becomes a controllable disease, conventional BCG maintenance therapy might help decrease the recurrence rate of NMIBC. Studies of patients receiving 1-year maintenance therapy should be conducted in the near future, and preparations for another pandemic should be made.

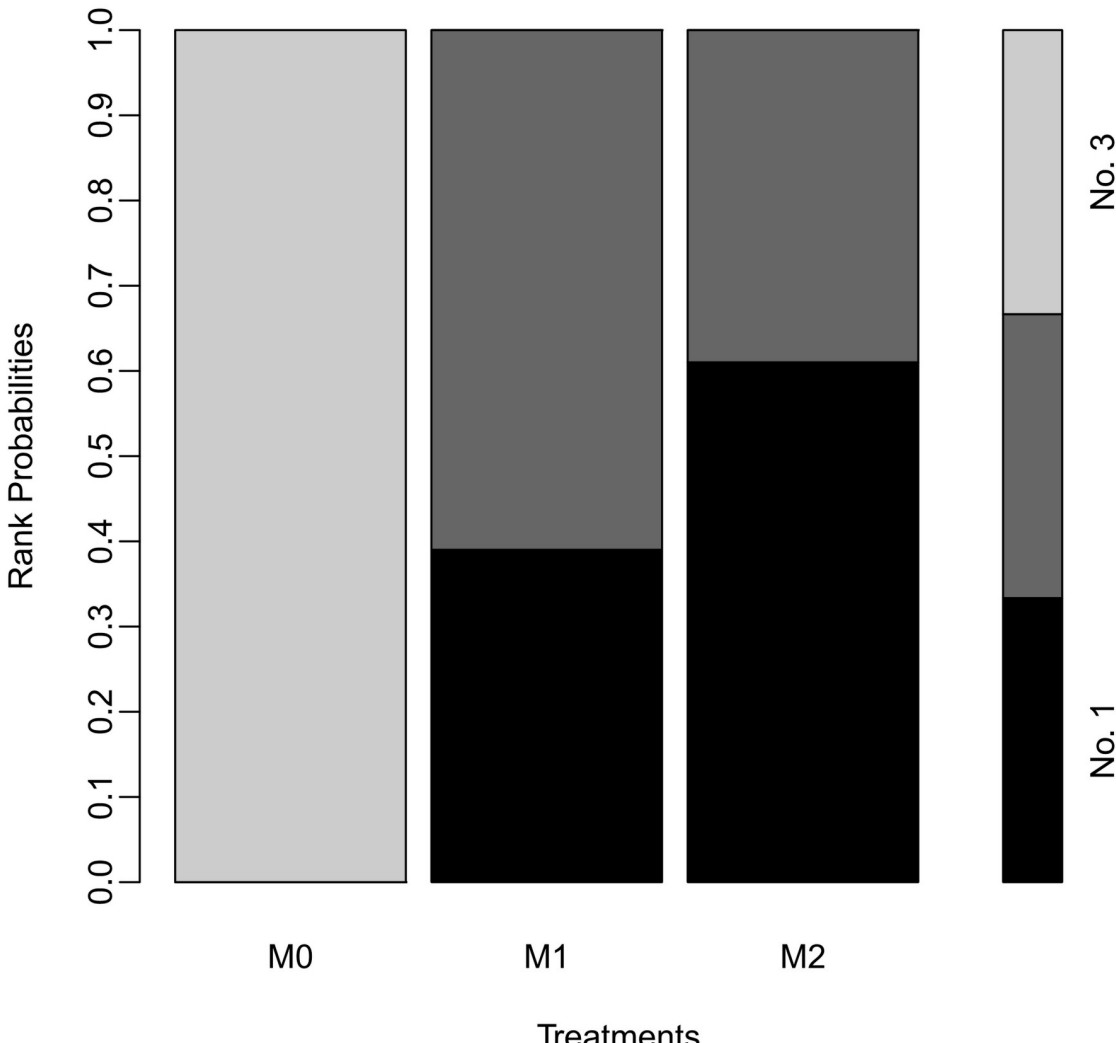

**Fig 10. The rank-probability test using MCMC modeling.** M0, induction BCG therapy only; M1, 1-year BCG maintenance therapy; M2, BCG maintenance therapy for more than 1 year.

## Supporting information

**S1 Table. PRISMA checklist.**
(DOCX)

## Author Contributions

**Conceptualization:** Joo Yong Lee.

**Data curation:** Young Joon Moon, Jae Yong Jeong.

**Formal analysis:** Doo Yong Chung, Hae Do Jung.

**Investigation:** Young Joon Moon, Jae Yong Jeong.

**Methodology:** Kang Su Cho, Joo Yong Lee.

**Project administration:** Joo Yong Lee.

**Resources:** Jae Yong Jeong, Joo Yong Lee.

**Software:** Doo Yong Chung, Hae Do Jung.

**Supervision:** Joo Yong Lee.

**Validation:** Dong Hyuk Kang, Joo Yong Lee.

**Visualization:** Doo Yong Chung, Dong Hyuk Kang.

**Writing – original draft:** Young Joon Moon.

**Writing – review & editing:** Kang Su Cho, Joo Yong Lee.

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
