## [Decision Letter · Decision Letter 0]

8 Jul 2022

PONE-D-22-09016Effects of intravesical BCG maintenance therapy duration on recurrence rate in high-risk non-muscle invasive bladder cancer (NMIBC): Systematic review and network meta-analysis according to EAU COVID-19 recommendationsPLOS ONE

Dear Dr. Lee,

Thank you for submitting your manuscript to PLOS ONE. After careful consideration, we feel that it has merit but does not fully meet PLOS ONE’s publication criteria as it currently stands. Therefore, we invite you to submit a revised version of the manuscript that addresses the points raised during the review process.

We look forward to receiving your revised manuscript.

Kind regards,

Giuseppe Lucarelli, M.D., Ph.D.

Academic Editor

PLOS ONE

Journal Requirements:

Reviewers' comments:

Reviewer's Responses to Questions

**Comments to the Author**

1. Is the manuscript technically sound, and do the data support the conclusions?

Reviewer #1: Yes

Reviewer #2: Yes

2. Has the statistical analysis been performed appropriately and rigorously? 

Reviewer #1: Yes

Reviewer #2: Yes

3. Have the authors made all data underlying the findings in their manuscript fully available?

Reviewer #1: Yes

Reviewer #2: No

4. Is the manuscript presented in an intelligible fashion and written in standard English?

Reviewer #1: Yes

Reviewer #2: Yes

5. Review Comments to the Author

Reviewer #1: Objectives: Authors purposes in this manuscript were to evaluate as a systematic and a network meta-analysis the recurrence rates after TURBT and BCG instillations (in patients with NMIBC receiving induction and maintenance therapy, under and over one year) available in the literature. The work has value in the context of identifying the best recommendation period of BCG instillation for NMIBC.

There are few minor issues that can improve the manuscript.

1. At line 72-73 the purpose of the review is indicated to be the appropriate duration of BCG maintenance therapy both during and after control of the COVID-19 pandemic. It is somehow misleading, because the readers can expect to see a comparative analysis on the studies performed pre and after pandemic. “After control of the COVID-19 pandemic” is the term that has to be modified. The Covid-19 pandemic is not over yet. I would like to ask the authors to rephrase the paragraph.

2. In the Discussion section, from line 194 to 199, the information can be better inserted in the introduction section.

Reviewer #2: I appreciate this meta-analysis and I only suggest minor revision as:

Introduction:

Add impact of reTurb on bcg response : see Urol Oncol. 2022 Jun 5:S1078-1439(22)00186-7. doi: 10.1016/j.urolonc.2022.05.016. Online ahead of print. ;

J Cancer 2018 Oct 20;9(22):4250-4254. doi: 10.7150/jca.26129. eCollection 2018.

Define concept of BCG unresposnsive , according new definition of FDA and add recent evidence of literature : Urol Oncol. 2022 Jun 5:S1078-1439(22)00186-7. doi: 10.1016/j.urolonc.2022.05.016. Online ahead of print.

In discussion remark the concept of delayed treatment during pandemic era

6. PLOS authors have the option to publish the peer review history of their article (what does this mean?). If published, this will include your full peer review and any attached files.

Reviewer #1: **Yes: **Tataru Octavian Sabin

Reviewer #2: **Yes: **matteo Ferro

---

## [Author Response · Author response to Decision Letter 0]

12 Aug 2022

● Reviewer #1:

Comment 1-1: At line 72-73 the purpose of the review is indicated to be the appropriate duration of BCG maintenance therapy both during and after control of the COVID-19 pandemic. It is somehow misleading, because the readers can expect to see a comparative analysis on the studies performed pre and after pandemic. “After control of the COVID-19 pandemic” is the term that has to be modified. The Covid-19 pandemic is not over yet. I would like to ask the authors to rephrase the paragraph.

Answer 1-1: We revised the paragraph according to your comment.

Therefore, the purpose of this study was to determine the appropriate duration of BCG maintenance therapy during the COVID-19 pandemic since 2020.

Comment 1-2: In the Discussion section, from line 194 to 199, the information can be better inserted in the introduction section.

Answer A-2: According to your comment, we inserted the paragraph in the introduction section.

 

● Reviewer 2:

Comment 2-1: Add impact of reTurb on bcg response : see Urol Oncol. 2022 Jun 5:S1078-1439(22)00186-7. and J Cancer 2018 Oct 20;9(22):4250-4254. 

Answer 2-1: As you commented, we added the reference as follows.

Ferro et al. conducted a retrospective study to investigate the predictive factors in the response to BCG in patients with a T1G3/HG NMIBC diagnosis [42]. According to their study, multifocality, lymphovascular invasion, and HG on re-TURBT were independent predictors for response to BCG treatment. To reduce the risk of understaging and missing MIBC, re-TURBT should be performed, especially in HG NMIBC. According to another study, independent predictors to identify patients at risk of residual HG disease after a complete TURBT include the tumor size, presence of CIS, and body mass index (BMI)�25 kg/m2 [43]. When deciding to perform re-TURBT, the presence of these factors would also be an important consideration.

Comment 2-2: Define concept of BCG unresposnsive , according new definition of FDA and add recent evidence of literature : Urol Oncol. 2022 Jun 5:S1078-1439(22)00186-7. doi: 10.1016/j.urolonc.2022.05.016. Online ahead of print.\\

Answer 2-2: According to your comment, we added the reference as follows.

BCG unresponsiveness is one of the important considerations in BCG therapy for patients with NMIBC. According to the EAU guidelines on NMIBC, BCG unresponsive tumors included all BCG refractory tumors (T1G3/high-grade (HG) tumor at 3 months; TaG3/HG tumor after 3 months and/or at 6 months, after either re-induction or first course of maintenance; carcinoma in situ (CIS), without concomitant papillary tumor, at 3 months and persisting at 6 months after either re-induction or first course of maintenance; HG tumor during BCG maintenance therapy) and those who develop T1Ta/HG recurrence within 6 months or CIS within 12 months from the completion of adequate BCG exposure [41]. In general, BCG-unresponsive patients have worse oncological outcomes, and therefore, studies related to factors that can predict BCG response are important in the treatment of NMIBC.

Comment 2-3: In discussion remark the concept of delayed treatment during pandemic era.

Answer 2-3: We added the reference according to your comment.

During the COVID-19 pandemic, management also has been delayed for muscle-invasive bladder cancer (MIBC). According to EAU recommendations, Kang et al. conducted systematic review and meta-analysis to evaluate the efficacy of neoadjuvant chemotherapy (NAC) compared with radical cystectomy (RC) alone in improving the overall survival (OS) of patients with T2-4aN0M0 MIBC. The OS was significantly better in the NAC with RC group than in the RC alone group. However, in a subgroup analysis of patients with T2N0M0 MIBC, there was no difference in the OS between the NAC with RC group and the RC alone group. They concluded that, as recommended by the EAU Guidelines Office Rapid Reaction Group, patients with T2N0M0 MIBC should strongly consider omitting NAC until the end of the COVID-19 pandemic.

---

## [Editor Report · Decision Letter 1]

15 Aug 2022

Effects of intravesical BCG maintenance therapy duration on recurrence rate in high-risk non-muscle invasive bladder cancer (NMIBC): Systematic review and network meta-analysis according to EAU COVID-19 recommendations

PONE-D-22-09016R1

Dear Dr. Lee,

We’re pleased to inform you that your manuscript has been judged scientifically suitable for publication and will be formally accepted for publication once it meets all outstanding technical requirements.

Kind regards,

Giuseppe Lucarelli, M.D., Ph.D.

Academic Editor

PLOS ONE
---

## [Editor Report · Acceptance letter]

30 Aug 2022

PONE-D-22-09016R1 

Effects of intravesical BCG maintenance therapy duration on recurrence rate in high-risk non-muscle invasive bladder cancer (NMIBC): Systematic review and network meta-analysis according to EAU COVID-19 recommendations

Dear Dr. Lee:

I'm pleased to inform you that your manuscript has been deemed suitable for publication in PLOS ONE. Congratulations! Your manuscript is now with our production department. 

Kind regards, 

on behalf of

Dr. Giuseppe Lucarelli 

Academic Editor

PLOS ONE